# Towards Finding Longer Proofs

## Abstract

We present a reinforcement learning (RL) based guidance system for automated theorem proving geared towards Finding Longer Proofs (FLoP). FLoP is a step towards learning to reason by analogy, reducing the dependence on large scale search in automated theorem provers. We use several simple, structured datasets with very long proofs to show that FLoP can successfully generalise a single training proof to a large class of related problems, implementing a simple form of analogical reasoning. On these benchmarks, FLoP is competitive with strong theorem provers despite using very limited search.

## 1 Introduction

Automated Theorem Proving (ATP) is the study of using machines for formal mathematical reasoning. It is related to general game playing, for example, the game of Go can be viewed as a simple formal system. Building on the recent success of machine learning, a growing trend in this field is to use learning methods to make theorem provers more powerful. Several research projects have shown that learning can be used to replace/surpass human-engineered heuristics. Despite huge improvements, interesting mathematical theorems remain elusive today. One crucial shortcoming of ATP systems is that they can typically find only relatively short proofs.

In this paper, we address this shortcoming and ask the question of how machine learning can be used to solve problems requiring very long inference chains. We argue that the fundamental reason why current ATP systems are limited to short proofs is that they focus on the search aspect of the task. It is very natural to see theorem proving as a search problem: each proof step involves a choice from a set of valid inferences, yielding a search space that grows exponentially with the length of the proof. Due to the exponential blowup, the search is bound to fail beyond a certain depth – except for special classes of problems where one of the smart human heuristics of the theorem prover allows for finding the solution without a search. As W. W. Bledsoe observed (Bledsoe, 1986): "Automated theorem proving ... is not the beautiful process we know as mathematics. This is 'cover your eyes with blinders and hunt through a cornfield for a diamond-shaped grain of corn'."

Approaches that try to avoid excessive search broadly fall into three categories: 1) Perform large steps, such as the invocation of tactics or decision procedures in SMT solvers (Barrett & Tinelli, 2018). 2) Perform hierarchical reasoning by first creating a high-level proof plan and then gradually refine it to the calculus level, e.g. Bundy (1988); Melis & Siekmann (1999). 3) Reason by analogy, e.g. Melis (1995); Brock et al. (1988).

Reasoning by analogy involves observing the proof of one problem, extracting the core idea, and successfully applying it to another. Note that using this formulation, success is barely dependent on proof length. On the other hand, establishing mappings between proofs is challenging and depends heavily on a proper data representation, which has been from the beginnings of ATP a major bottleneck for this approach. However, with the advent of machine learning methods capable of automatically discovering good data embeddings, the analogy approach seems worth revisiting. Our work aims to identify machine learning methods that are suitable for analogical reasoning, and as a result capable of solving problems with long proofs.

Many successful ATP systems (Urban et al., 2008; Jakubuv & Urban, 2019; Chvalovský et al., 2019; Bansal et al., 2019a; Kaliszyk et al., 2018; Zombori et al., 2020; Olšák et al., 2020; Polu & Sutskever, 2020) implement the MaLARea Urban (2007); Urban et al. (2008) learning/reasoning loop (described later also as the DAgger Ross et al. (2011) meta-algorithm). The MaLARea loop interleaves ATP

runs based on the current models (*data collection phase*) with a *training phase*, in which these models are updated to fit the collected data.

An alternative family of reinforcement learning methods, including Temporal Difference (TD) learning (Sutton & Barto, 2018), continuously update their models, allowing the system to bootstrap on itself. Such methods have so far been mostly ignored by the theorem proving community. In these methods, the search is usually replaced by rollouts. Our paper argues that while the DAgger approach is more suitable to learn good search heuristics, methods with strong bootstrapping are better at learning to reason by analogy.

Our work has the following contributions.

1. We introduce a new theorem proving algorithm FLoP (Section 3) based on a TD algorithm [1] and the connection tableau calculus. FLoP makes use of a curriculum learning algorithms presented by Resnick et al. (2018) and Salimans & Chen (2018). These techniques are well established in RL, however, they have never been applied to theorem proving before.
2. We introduce a synthetic dataset of increasingly difficult arithmetic problems, as well as two datasets from the Logical Calculi domain of the TPTP (Sutcliffe, 2017) library, augmented with lemmata (Section 4).
3. We show that when restricted to single shot evaluation – without search – FLoP performs very well, while another prover based on guided Monte Carlo Tree Search greatly degrades.
4. We evaluate FLoP on our arithmetic benchmarks by training it on a single problem and show that it generalizes very well even when evaluated without search, allowing just a few proof attempts. This suggests that it has learned a simple form of reasoning by analogy.
5. We use the arithmetic benchmarks to compare FLoP with state-of-the-art provers Vampire (Kovács & Voronkov, 2013), E (Schulz, 2013), leanCoP (Otten & Bibel, 2003) guided by human-designed strategies, and with rlCoP (Kaliszyk et al., 2018) – an RL-based connection tableau prover. In the simple setup of unary encoding of numbers, FLoP is only outperformed by a portfolio (multi-strategy) mode of a single manually optimized rewriting-based system and only after trying several of its autoconfiguration heuristics. When using binary encoding, FLoP performs best, demonstrating its ability to generalize to long proofs.

Our datasets presented in Section 4 seem to be particularly suited for machine learning methods: some problems are algorithmically simple, with long solutions and strong shared structure (Robinson Arithmetic) while others are less similar, but hierarchically structured (Logical Calculi). Nevertheless, state-of-the-art systems struggle with solving some of the problems (see Section 6). Furthermore, our problems are much easier to analyze than typical heterogeneous proof corpora. We claim that our datasets are of the very few that not only challenge theorem provers but also allow for understanding their current limits. The difficulty of our synthetic problems, as well as their proof length, are easily adjustable, yielding a scalable RL benchmark with interpretable failure modes (see Appendix I).

Our code, datasets and all experiment configuration files are available at `http://bit.ly/code_atpcurr`[2]. Supplementary materials including screencasts with gameplays performed in our environments are available at the project webpage `http://bit.ly/site_atpcurr`.

## 2 THEOREM PROVING BY ANALOGY

Analogy has long been considered one of the most important heuristics in mathematical problem solving, e.g. Polya (1971). It also gained attention in automated theorem proving, e.g.Brock et al. (1988); Melis (1995), as an alternative of search-based methods.

Brock et al. (1988) define analogical reasoning as "the proof of one theorem is used to guide the proof of a similar theorem by suggesting analogous steps". They rely on a user-provided matching between analogous concepts related to the two theorems and try to reuse the proof steps (adjusted modulo analogy) in the source proof during the construction of the target. Melis (1995) aim to achieve this on a higher level of abstraction by matching proof plans of a source and a target problem. As the

---

[1] In particular, we use Proximal Policy Optimization (Schulman et al., 2017) (PPO), a variant of the policy gradient method, which uses Temporal Difference learning for optimization of the value function.

[2] This distribution does not include the fCoP theorem prover, which cannot yet be publicly released, however, a binary can be obtained upon request.

proof plan of the target problem is constructed, the plan of the source is searched for steps that can be transformed into a suitable step for the target. The set of allowable transformations are predefined and designed for a narrow domain. For example, the transformations given in Melis (1995) aim to carry a result, such as the Heine Borel theorem, stated in $\mathbb{R}^1$ over to $\mathbb{R}^2$. The characteristic feature of these systems is that search is performed on the meta level of plan mappings and proof step transformations, which is often a much smaller search space than that defined by the inference rules of the calculus.

A machine learning system that is trained to guide a theorem prover is supposed to achieve a similar result, with two important improvements. First transformations are learned, without the need for manual engineering. Second, establishing mappings between proof steps (that can be transformed into each other) should result from learning of flexible and abstract features. The flexibility and abstraction allows for potential reusing the same proof components several times, as well as using components from different proofs, which goes beyond earlier attempts that only establish direct matching between the two proofs.

Despite the unquestioned role of analogical reasoning in human thinking, modern theorem provers have abandoned this direction and focused on developing better heuristics and architectures to support calculus level proof search. Machine learning is well suited for capturing similarities and analogies. In this paper, we intend to point out the attractive aspect of analogical reasoning in the context of machine learning and provide datasets and benchmarks which enforce very long proofs and in turn make the search very hard, emphasizing a need for reasoning via analogy.

## 3  FLoP – Main Algorithm

FLoP combines the connection tableau calculus with guidance based on Temporal Difference and curriculum learning. A brief introduction to the connection tableau calculus is provided in Appendix J. After each inference step, the prover engine returns its current state as well as the set of valid actions. Figures 4 and 1 show how actions interact with other components of FLoP. The state and actions (formulae) are represented using previously developed features Kaliszyk et al. (2015a), described in Appendix F. Algorithm 1 gives an overview of the learning loop.

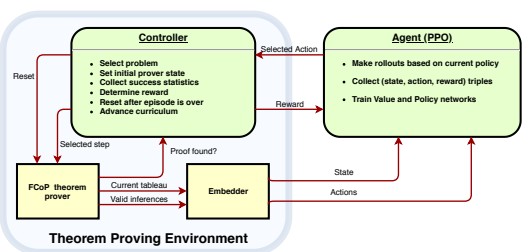

Figure 1: *Theorem proving as a reinforcement learning environment.*

First, in line 5 we sample a problem. In lines 6–9 we interact with the prover and ensure that its state corresponds to the one dictated by the current curriculum. In lines 10–15 we generate a rollout iterating 1) prover steps, 2) featurization and 3) sampling a next action according to the policy. If a new problem is solved, we start the curriculum on it in lines 17–19. If performance on a given problem and curriculum reaches a threshold, we advance the curriculum in lines 20–22. In line 24 we update the policy and value models based on the specified number of episodes.

## 4  Datasets

To evaluate our system, we select simple classes of theorems with strong shared structure, giving a large room for learning-based improvement. Our five datasets are described in Table 1. The datasets are bundled into an OpenAI-gym (Brockman et al., 2016) compliant environment and can be tested with modern RL algorithms.

Three datasets are built on the theory of Robinson Arithmetic (Robinson, 1950), which defines addition and multiplication on the nonnegative integers. Despite its relative simplicity, this theory seems to be particularly suited for machine learning methods: solutions are long and repetitive, while also challenging for state-of-the-art systems (Section 6). We examine both unary (24 actions) and binary (40 actions) encoding of numbers. The axioms of Robinson Arithmetic are given in Appendix B and C.

Two datasets are extracted from the TPTP library, from the domain of Logical Calculi with condensed detachment (LCL). These theorems have been extensively studied from the early days of automated

---

**Algorithm 1** FLoP: Main Learning Loop

---

**Require:** problems $\mathcal{P}$, policy $\pi$, value $v$, train steps $\in \mathbb{N}$, threshold $\in [0..1]$, episodes between updates: $k \in \mathbb{N}$
**Ensure:** trained policy $\pi$, trained value $v$, possibly proofs for some problems in $\mathcal{P}$
1: $curriculum \leftarrow$ dictionary such that for each $p \in P$ with proof $Pr$ $curriculum[p] = len[Pr] - 1$
2: steps $\leftarrow 0$
3: **while** steps < train steps **do**
4:    **for** j in 1..k **do**
5:       $p \leftarrow$ random problem from problem set $\mathcal{P}$ {An episode corresponds to a problem}
6:       initialize prover on problem $p$
7:       **if** $p$ has stored proof **then**
8:          Take $curriculum[p]$ proof steps according to stored proof
9:       **end if**
10:      **while** not episode over **do**
11:         $s', a'_1, a'_2 \ldots a'_l \leftarrow$ Query prover for current state and valid actions
12:         $s, a_1, a_2 \ldots a_l \leftarrow \text{feat}(s'), \text{feat}(a'_1), \text{feat}(a'_2) \ldots \text{feat}(a'_l)$ {Extract features}
13:         Take action according to policy $\pi(a|s)$, observe reward $r$
14:         steps $\leftarrow$ steps + 1
15:      **end while**
16:      update success ratio for $p$
17:      **if** $p$ is solved with proof $Pr$ and no proof of $p$ was known before **then**
18:         $curriculum[p] \leftarrow len(Pr) - 1$ {Start curriculum}
19:      **end if**
20:      **if** success rate for $p$ > threshold **then**
21:         $curriculum[p] \leftarrow curriculum[p] - 1$ {Advance curriculum}
22:      **end if**
23:    **end for**
24:    Update policy $\pi$ and value $v$
25: **end while**

---

Table 1: *Three challenges defined in the theory of Robinson Arithmetic (RA) and two challenges from the Logical Calculi (LCL) domain of the TPTP library*

| Name | Theory | Size | Description |
|------|--------|------|-------------|
| **RA-1** | RA | 1800 | Expressions of the form $N_1 + N_2 = N_3$, $N_1 \cdot N_2 = N_3$, where $0 \le N_i < 30$. (Examples: $3+4=7$ or $5 \cdot 12=60$.) |
| **RA-2** | RA | 1000 | $T = N$, where $0 \le N$, and $T$ is a random expression with 3 operators and operands $N_i$ such that $0 \le N_i < 10$. (E.g.: $((3+4) \cdot 2)+6=20$.) |
| **RA-3** | RA | 1000 | $T_1 = T_2$, where $T_1$ and $T_2$ are random expressions with 3 operators and operands $N_i$ such that $2 \le N_i < 10$. E.g. $((3+4) \cdot 2)+6 = ((1+1) \cdot 5) \cdot 2$.) |
| **LCL-Eq** | LCL | 890 | TPTP domain: Logic Calculi (Equivalential) – extended with lemmata from E prover. |
| **LCL-Imp** | LCL | 1204 | TPTP domain: Logic Calculi (Implication/Falsehood 2 valued sentential) – extended with lemmata from E prover. |

theorem proving, e.g. McCune & Wos (1992); Peterson (1976); Kalman (1978); Wos (1990). We run E prover with a large time limit on the problems and augment the dataset with lemmata extracted by E. As a result, many proofs of simpler problems can be directly used as parts of the proofs of harder problems. A direct analogy from one problem to the other is usually not possible, however, limited search is often sufficient to connect the proofs of easier problems to the proof of harder ones. More details about the LCL datasets are given in Appendix D.

## 5 RELATED WORK

**Machine learning systems for guiding theorem provers.** A large body of research exists that aims to provide guidance for theorem provers via machine learning. FEMaLeCoP (Kaliszyk & Urban,

2015), rlCoP (Kaliszyk et al., 2018; Olsák et al., 2020) and plCoP (Zombori et al., 2020) guide the leanCoP (Otten & Bibel, 2003) compact connection tableau prover, which is also the system guided in our project. Loos et al. (2017); Jakubuv & Urban (2017); Chvalovský et al. (2019); Jakubuv et al. (2020) add learning-based guidance to the saturation based E prover (Schulz, 2013). The HOList project (Bansal et al., 2019b; Paliwal et al., 2019; Bansal et al., 2019a) builds guidance on the tactic level[3] for the HOL Light Harrison (1996) higher-order theorem prover. A distinctive feature of all these systems is that they rely heavily on an external search procedure, such as Monte Carlo Tree Search (Kocsis & Szepesvári, 2006), or the search engine of the guided prover. Learning is aimed at making search more efficient and it is implemented in alternating iterations of proof search and model fitting, according to the DAgger (Ross et al., 2011) meta-algorithm, first used in MaLARea (Urban et al., 2008) for theorem proving. In contrast with the above, we use an algorithm which emphasizes bootstrapping, aiming to guide the leanCoP prover with very limited search, focusing on learning an analogy between proofs of related problems.

Recently, several works Piotrowski & Urban (2020); Urban & Jakubuv (2020); Polu & Sutskever (2020) have used RNNs, attention and transformers to generate next proof steps. E.g., Polu & Sutskever (2020) reports generalisation on problems with relatively short proofs. In line with emphasizing analogy over search, the evaluation protocol used in Polu & Sutskever (2020) allows for only a very limited search in a single proof attempt [4]. Interestingly, they get a significant improvement (from 42.9% of theorems proved to 56.5% of theorems proved) by simply increasing the number of proof attempts from 2 to 32, which is a bigger gain then what is obtained via pretraining. Our work employs much smaller neural models and focuses on generalizing to proofs with hundreds and thousands of steps (see Figures 2 and 3).

**Provers guiding the leanCoP Connection Tableau Calculus.**     As noted above, a series of learning systems guide the leanCoP connection calculus. Of these, we highlight three systems that use roughly the same learning setup: rlCoP Kaliszyk et al. (2018), plCoP Zombori et al. (2020) and graphCoP Olsák et al. (2020). In these systems, the value and policy functions of the guided MCTS algorithm are learned similarly to (Anthony et al., 2017; Silver et al., 2017). FLoP shares the same manually developed features (Kaliszyk et al., 2015a) with rlCoP and plCoP, while graphCoP employs a graph neural network for feature extraction. We use these systems as an important baseline in Section 6. While the differences are important, they play little role in our current investigation and we refer to them jointly as *mcts-CoP*s

**Machine learning datasets and RL environments involving mathematics and logic.**     The TPTP dataset (Sutcliffe, 2017) consists of 22507 problems in 53 domains collected over several decades. A large dataset for developing machine learning for theorem proving based on the Mizar Mathematical Library (MML) (Grabowski et al., 2010) was introduced by Urban (2006a) in the MPTP project. Similar datasets based on the Isabelle/HOL, HOL Light/Flyspeck and HOL4/CakeML systems and projects (Blanchette et al., 2016; Kaliszyk & Urban, 2014; Gauthier & Kaliszyk, 2015) were introduced in the last decade and used for the CASC LTB (large theory) ATP competition (Sutcliffe & Urban, 2016) and other system evaluations. Such datasets cover large areas of mathematics and computer science and contain diverse axioms, lemmas, theorems, definitions, and symbols. Smaller subsets of lemmas leading to the Bolzano-Weierstrass theorem were selected from the MPTP dataset to form the MPTP Challenge (Urban, 2006b) and the MPTP2078 benchmark (Alama et al., 2014). HOLStep (Kaliszyk et al., 2017) introduced a dataset based on $11400$ proofs, including a proof of the Kepler Conjecture (Hales et al., 2015), formalized using HOL Light (Harrison, 1996). The HOList project (Bansal et al., 2019b; Paliwal et al., 2019; Bansal et al., 2019a) uses $29462$ theorems formalized in HOL Light and instruments them for experiments oriented towards tactic selection, where a tactic is a human-designed program which aggregates multiple proof steps. GamePad (Huang et al., 2019) introduced a dataset based on a formalization of the Feit-Thompson Theorem (Gonthier et al., 2013) along with generated algebra problems. It is intended for learning tactic selection together with an auxiliary task of predicting the number of proof steps left. A dataset based on theorems proved in HOL4 (Slind & Norrish, 2008) was used for developing the TacticToe (Gauthier et al., 2018) learning-guided tactical prover. Simple arithmetic equalities are used in the evaluation of GPT-3 Brown et al. (2020). Their results indicate that generalization in this domain is hard

---

[3]A tactic is a human-designed program which aggregates multiple proof steps.

[4]A maximum of 4096 search nodes are allowed.

beyond a few digits even for powerful language models using billions of parameters. The inequality benchmark introduced in Wu et al. (2020) allows for generating and proving arithmetic inequalities. The benchmark is not intended to deal with long proofs and is not related to other domains of mathematics.

The datasets which we introduce in Section 4 are structurally much simpler than most other theorem proving datasets. Our arithmetic dataset contains problems that require extremely long proofs, however, the proofs are structurally similar. The dataset about logical calculi contains various challenge problems from the TPTP library, extended with lemmata. The added lemmata make the dataset hierarchical: proofs of simpler lemmata can be directly incorporated into harder proofs. We argue that these datasets are suitably challenging for the current learning methods and are intended to become a general-purpose testing ground for theorem proving and reinforcement learning methods, much like grid worlds for RL. All arithmetic problems in our dataset are quite simple for humans, but in the case of logical calculi, some of the problems in the dataset were posing a challenge for mathematicians (see Wos et al. (1984)).

## 6 EXPERIMENTS

Our experiments with Robinson arithmetic aim to demonstrate that in this highly structured dataset FLoP is capable of extracting a general proof pattern from one or two proofs and successfully generalizing to related proofs of arbitrary length, using a restricted few-shot evaluation method (see below). Experiments 1, 2, and 3 compare FLoP with strong theorem provers using different fragments of the arithmetic dataset, varying the complexity of the axiomatization (unary vs. binary encoding of numbers) and the complexity of the target theorems (RA-1, RA-2, RA-3). We find that FLoP is either the best or the second-best in each experiment. In each of these experiments, FLoP is allowed 100 proof attempts without backtracking:

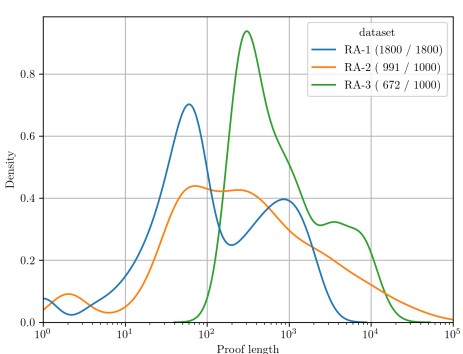

Figure 2: *Distributions of length of proofs found by* FLoP *on RA-1, RA-2, RA-3. Average proof lengths are 367, 2082, and 1864, respectively. Note the logarithmic scale.*

ing: the first attempt is a deterministic run with a high time limit (10000 sec) that always selects the action maximizing the policy and the remaining 99 runs are stochastic samples from the policy with a time limit of 60 sec.

The LCL problems used in our experiments are less structured and success is dependent on search, even if the hierarchical composition of problems ensures that a relatively small search is sufficient to generalize from easier problems to harder ones. Consequently, we expect that search-based methods are better in this domain. However, when search is completely disallowed during evaluation, we show in Experiment 4 that FLoP performs much better than the mcts-CoPs. Finally, in Experiment 5 we demonstrate the benefit of using curriculum learning.

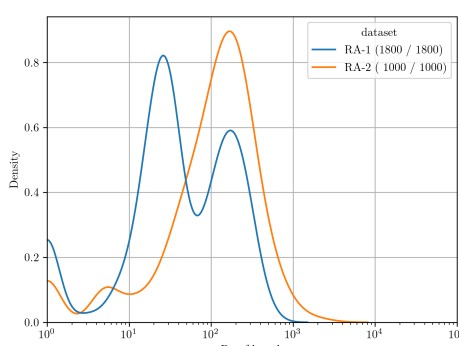

Figure 3: *Distributions of length of proofs found by* FLoP *on binary RA-1, binary RA-2. Average proof lengths are 85 and 179. Note the logarithmic scale.*

**Experiment 1: Comparison with other provers.** We compare FLoP with a random model, two state-of-the-art saturation-style theorem provers (E, Vampire), a heuristic guided connection tableau prover (leanCoP), and rl-CoP (one of the mcts-CoPs). Vampire, E, and leanCoP use human-designed strategies instead of learning. We use these provers in the configuration used for CASC, the yearly competition of fully automated theorem provers, employing a time limit of 60 sec. per problem. For E, we also report the

results of the *auto-schedule* mode. For rlCoP we used the hyperparameters described in Kaliszyk et al. (2018), only modifying the policy temperature from 2.5 to 1.5, as this works better with the Robinson datasets. The number of inferences in MCTS was limited to 200000. rlCoP was trained on the whole evaluation set, while FLoP was trained on a single problem: $1 \cdot 1 = 1$ and $1 \cdot 1 \cdot 1 = 1$ for RA-1 and RA-2, respectively.[5]

Success ratios are given in Table 2. FLoP is only outperformed by E's *auto-schedule*, which tries multiple strategies and finds one with the left-to-right ordering of all the addition and multiplication axioms. This solves all of our problems immediately without proof search by only rewriting to a normal form (Baader & Nipkow, 1998). This demonstrates the power of equational theorem proving when a suitable term ordering exists and can be found by human-designed heuristics.

Table 2: *Comparing a random model, Vampire, E, leanCoP, rlCoP and FLoP, with respect to success ratio for RA-1, RA-2 and RA-3 problems. Our method (FLoP) is marked in grey. $E_1$ – auto mode, $E_2$ – auto-schedule mode, $E_3$ – auto-schedule with renamed equality. The reason why FLoP did not reach 100% on RA-2 is that a few problems time-outed.*

| Dataset | Random | Vampire | $E_1$ | $E_2$ | $E_3$ | leanCoP | rlCoP | FLoP |
|---------|--------|---------|-------|-------|-------|---------|-------|------|
| RA-1 | 0.04 | 0.60 | 0.60 | **1.0** | 0.54 | 0.22 | 0.86 | **1.0** |
| RA-2 | 0.05 | 0.40 | 0.39 | **1.0** | 0.25 | 0.14 | 0.74 | 0.99 |
| RA-3 | 0.00 | 0.34 | 0.28 | **1.0** | 0.22 | 0.01 | 0.41 | 0.67 |

This is, however, far from guaranteed in general even in such simple domains, as witnessed by Vampire's failure to find this ordering. To evaluate E without access to its built-in rewriting capability, we have renamed the equality to a new predicate 'eq' axiomatized exactly in the same way as in leanCoP. The auto-schedule mode then becomes somewhat weaker than the auto mode, see Table 2.

**Experiment 2: Harder Arithmetic Expressions.** RA-3 consists of arithmetic equalities with random expressions on both sides. This dataset is significantly more complex because there are many ways of proving the same problem. Proofs are longer, too. For FLoP, we examined various training sets and found that the system is very prone to overfitting. Most problems can be proven in many different ways, that vary greatly in terms of how well they foster generalization. It is true especially of easier problems that they can be proven with "shortcuts" that hinder generalization (See more on this in Appendix I). The harder the problems, the less likely they can be solved with such heuristic approaches, hence harder training problems promise better training signal. We demonstrate this by training FLoP on a few harder problems with proofs provided, making use of curriculum learning described in Section 3. A single longer training proof is sufficient to yield meaningful generalization. Adding one more training problem helps even more, as shows Table 3.

Table 3: *Curriculum learning for RA-3 on two harder problems with proofs of 113 and 108 steps. We report success ratios and average proof lengths, based on 3 runs. Standard deviations are given in parenthesis.*

| Training problem | Succ. | Len. |
|------------------|-------|------|
| $1 \cdot 2 + 1 + 1 = (1 + 1) \cdot 1 \cdot 2$ | 0.32(0.05) | 566(14) |
| $1 \cdot 2 + 1 + 1 = (1 + 1) \cdot 1 \cdot 2$ $(1 + 1 + 1) \cdot 2 = 2 \cdot 1 + 2 + 2$ | **0.67** (0.03) | 1864(54) |

Figure 2 shows the distribution of the length of proofs found by FLoP. We can see that a large part of the problems requires thousands of steps to solve, highlighting the need to avoid search.

For rlCoP, all RA-3 problems are too hard to solve without guidance within the inference limit, so we started with the version trained on the solutions of RA-2. Table 2 shows that FLoP is only outperformed by E's auto-schedule mode, which again finds the rewrite ordering that solves all problems without search.

**Experiment 3: Binary Number Encoding.** We experiment with Robinson Arithmetic using binary encoding of numbers. This makes the domain theory more complex: the total number of actions increases from 24 to 40. [6]. On the other hand, proofs get shorter, as shows Figure 3. Again, we train FLoP on a single proof:

Table 4: *Comparing Vampire, E (auto-schedule mode), leanCoP, rlCoP and FLoP, using binary encoding of numbers.*

| Dataset | Vampire | E | leanCoP | rlCoP | FLoP |
|---------|---------|------|---------|-------|------|
| RA-1 | 0.67 | 0.81 | 0.19 | 0.56 | **1.0** |
| RA-2 | 0.62 | 0.62 | 0.13 | 0.12 | **1.0** |

$3 \cdot 3 = 9$ and $(1 \cdot 2 + 1) \cdot 3 = 9$ for RA-1 and RA-2, respectively. Table 4 shows that provers get weaker, except for Vampire and FLoP. In particular, E is no longer capable of solving the problems with rewriting only. FLoP manages to generalize from a single proof to the whole dataset despite the increased action space and performs best in this experiment.

---

[5]For a description of RA-3 training problems, see Experiment 2.

[6]Note that only a subset of these is applicable in a given state.

**Experiment 4: Search vs. Eager Evaluation**
We compare FLoP with plCoP (one of the mcts-CoPs) using two different evaluation methods. After training both systems on the whole dataset, we evaluate them using 1) MCTS and 2) eager evaluation, i.e. always select the action with the highest probability according to the policy model. Table 5 shows that plCoP performs better when search is allowed, especially for the more heterogeneous LCL problems. However, FLoP takes the upper hand in eager evaluation.

Table 5: *Comparing FLoP and plCoP using two different evaluation methods: 1) guided MCTS and 2) eager evaluation based on the policy model*

| Dataset | Prover | MCTS | Eager Policy |
|---------|--------|------|--------------|
| LCL-Eq  | plCoP  | **47%** | 5% |
|         | FLoP   | 19% | **19%** |
| LCL-Imp | plCoP  | **61%** | 5% |
|         | FLoP   | 24% | **27%** |
| RA-1    | plCoP  | **65%** | 82% |
|         | FLoP   | 61% | **100%** |
| RA-2    | plCoP  | **48%** | 49% |
|         | FLoP   | 31% | **99%** |

For the LCL problems, plCoP collapses while FLoP is unaffected. This suggests that plCoP depends heavily on the search procedure it used for training. FLoP cannot make good use of MCTS, which is somewhat expected, since its policy and value networks were not trained for that purpose.

For the arithmetic datasets, both systems benefit from not doing search because they reach proofs that are longer than what MCTS can reach. For FLoP, the removal of the depth limit reveals that it fully mastered the two problem classes, regardless of depth. The performance of plCoP gets even worse if the eager evaluation is based on the value model, see Appendix H. These results are in line with our conjecture that the DAgger approach of plCoP is better for learning good search heuristics, while FLoP is better at analogical reasoning.

**Experiment 5: Curriculum Learning vs only Exploration Based Learning.** When training proofs are not available, the positive reward signal only occurs after the system solves a problem through exploration. Afterward, curriculum learning ensures that the system is continuously faced with a "reasonably" hard problem, alleviating the sparse reward problem of theorem proving.

Table 6: *Curriculum Learning compared with only exploration based learning, on the LCL-Eq and LCL-Imp datasets, using 10M and 30M inference limit, respectively. We report the ratio of proofs found during training. The results are averages of 2 runs.*

| Dataset | Curriculum | No curriculum |
|---------|------------|---------------|
| LCL-Eq  | **0.24** (0) | 0.23 (0.001) |
| LCL-Imp | **0.51** (0.002) | 0.45 (0.003) |

We demonstrate this on the two LCL datasets. Here, before generating each rollout, we randomly select a problem from the entire dataset. We report the number of proofs found during training in Table 6. Curriculum learning brings a small, but consistent improvement when compared with only exploration-based learning.

## 7 CONCLUSION AND FUTURE WORK

We have built FLoP, a proof guidance system based on a variant of temporal difference reinforcement learning, addressing the problem of finding long proofs in an exponential search space. Previous work (Veroff, 1996; Kinyon et al., 2013) focused on finding long proofs with the help of human-designed heuristics. We showed that FLoP is capable of extracting proof patterns via learning and can generalise to much longer proofs, implementing a simple form of reasoning by analogy. We believe that mastering analogical reasoning is an important step in creating human-level automated mathematicians. We presented a set of theorem proving datasets that are suitably challenging for existing learning methods and are intended to become a general-purpose testing ground for theorem proving and reinforcement learning methods. We showed that FLoP can outperform strong theorem provers on some of these datasets. We find that curriculum learning is a useful component of the learning algorithm as it allows for amplifying training signal when proofs are long.

In this paper, we focused on extracting a single proof pattern during training. A natural continuation will be to extract a portfolio of patterns from a larger pool of training problems. Transformer models are promising tools to achieve this, given their recent success in mastering several tasks in parallel. Transformers might also be capable of producing large chunks of proofs in a single inference step.

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

## A  FLoP Model Architecture

Figure 4: *Value and Policy network architectures in PPO. Their inputs are state and state-action pair features, respectively. The policy returns a score for each action, which are then normalized to a probability.*

Figure 4 shows the policy and value network architectures of the Proximal Policy Optimization (PPO) algorithm, as implemented in FLoP.

## B  Robinson Arithmetic

Robinson Arithmetic defines basic properties of arithmetic expressions on the nonnegative integers. The signature of the language contains atom '$o$' (representing 0), functions '$s$', 'plus' and 'mul' (for $+1$, $+$ and $\cdot$, respectively), and the equality predicate '$=$'. For example, formula $2 \cdot 1 + 1 = 3 + 0$ is written as

$$plus(mul(s(s(o)), s(o)), s(o)) = plus(s(s(s(o))), o).$$

We use the axioms provided in Table 7. The table also contains special axioms that are added by leanCoP to handle the equality predicate. The unary representation of numbers (e.g., $s(s(s(o)))$ represents 3) results in large expressions and long proofs as the numbers increase. For example, $((8 + 5) \cdot 8) \cdot 5 = 520$ takes over 16000 steps to prove in fCoP. We show an example of such proof on the project website. The total number of actions if 24.

## C  Robinson Arithmetic with Binary Encoding

Using binary representation makes the theory of Robinson Arithmetic more complex. Constant symbols $n0$ and $n1$ stand for 0 and 1, while term $b(X, Y)$ represents $X + 2 \cdot Y$. For example, 10 is represented as $b(n0, b(n1, b(n0, n1)))$. We can see in Table 8 that the number of axioms increases from $6 + 6$ to $17 + 6$. Correspondingly, the number of actions during proof search increases from 24 to 40.

Table 7: *Axioms of Robinson Arithmetic extended with special axioms for handling equality.*

| Name | Axiom |
|---|---|
| zero successor | $\forall X : \neg(o = s(X))$ |
| different successors | $\forall X, Y : (s(X) = s(Y)) \Rightarrow (X = Y)$ |
| plus zero | $\forall X : plus(X, o) = X$ |
| plus successor | $\forall X, Y : plus(X, s(Y)) = s(plus(X, Y))$ |
| mul zero | $\forall X : mul(X, o) = o$ |
| mul successor | $\forall X, Y : mul(X, s(Y)) = plus(mul(X, Y), X)$ |
| equality reflexivity | $\forall X : X = X$ |
| equality symmetry | $\forall X, Y : (X = Y) \Rightarrow (Y = X)$ |
| equality transitivity | $\forall X, Y, Z : (X = Y) \wedge (Y = Z) \Rightarrow (X = Z)$ |
| congruence of s | $\forall X, Y : (X = Y) \Rightarrow (s(X) = s(Y))$ |
| congruence of plus | $\forall X_1, X_2, Y_1, Y_2 : (X_1 = X_2) \wedge (Y_1 = Y_2) \Rightarrow plus(X_1, Y_1) = plus(X_2, Y_2)$ |
| congruence of mul | $\forall X_1, X_2, Y_1, Y_2 : (X_1 = X_2) \wedge (Y_1 = Y_2) \Rightarrow mul(X_1, Y_1) = mul(X_2, Y_2)$ |

Table 8: *Axioms of Robinson Arithmetic using binary encoding of numbers, extended with special axioms for handling equality.*

| Name | Axiom |
|---|---|
| zero successor | $\forall X, Y : \neg(n0 = b(X, Y))$ |
| one successor | $\forall X, Y : \neg(n1 = b(X, Y))$ |
| different successors | $\forall X_1, X_2, Y_1, Y2 : (b(X_1, Y_1) = b(X_2, Y_2)) \Rightarrow ((X_1 = X_2) \wedge (Y_1 = Y_2))$ |
| predecessor | $\forall X : (X = n0) \vee (X = n1) \vee (\exists Y, Z : b(Y, Z) = X)$ |
| plus zero | $\forall X : plus(X, n0) = n1$ |
| plus one1 | $plus(n0, n1) = n1$ |
| plus one2 | $plus(n1, n1) = b(n0, n1)$ |
| plus one3 | $\forall X : (plus(b(n0, X), n1) = b(n1, X))$ |
| plus one3 | $\forall X : (plus(b(n1, X), n1) = b(n0, plus(X, n1)))$ |
| plus more1 | $\forall X, Y : (plus(n0, b(X, Y)) = b(X, Y))$ |
| plus more2 | $\forall X, Y : (plus(n1, b(X, Y)) = plus(b(X, Y), n1))$ |
| plus more3 | $\forall X_1, Y_1, X_2, Y_2 : (plus(b(X_1, Y_1), b(X_2, Y_2)) = plus(b(X_1, plus(Y_1, Y_2)), X_2))$ |
| mul zero1 | $\forall X : (mul(X, n0) = n0)$ |
| mul zero2 | $\forall X : (mul(n0, X) = n0)$ |
| mul one1 | $\forall X : (mul(X, n1) = X)$ |
| mul one2 | $\forall X : (mul(n1, X) = X)$ |
| mul more | $\forall X_1, Y_1, X_2, Y_2 : (mul(b(X_1, Y_1), b(X_2, Y_2)) = plus(plus(plus(b(n0, b(n0, mul(Y_1, Y_2))), b(n0, mul(Y_1, X_2))), b(n0, mul(X_1, Y_2))), mul(X_1, X_2)))$ |
| equality reflexivity | $\forall X : X = X$ |
| equality symmetry | $\forall X, Y : (X = Y) \Rightarrow (Y = X)$ |
| equality transitivity | $\forall X, Y, Z : (X = Y) \wedge (Y = Z) \Rightarrow (X = Z)$ |
| congruence of b | $\forall X_1, X_2, Y_1, Y_2 : (X_1 = X_2) \wedge (Y_1 = Y_2) \Rightarrow (b(X_1, Y_2) = b(X_2, Y_2))$ |
| congruence of plus | $\forall X_1, X_2, Y_1, Y_2 : (X_1 = X_2) \wedge (Y_1 = Y_2) \Rightarrow plus(X_1, Y_1) = plus(X_2, Y_2)$ |
| congruence of mul | $\forall X_1, X_2, Y_1, Y_2 : (X_1 = X_2) \wedge (Y_1 = Y_2) \Rightarrow mul(X_1, Y_1) = mul(X_2, Y_2)$ |

## D  LCL DATASETS

The LCL domain in the TPTP (Sutcliffe, 2017) library consists of statements about various formal inference systems. LCL-Eq and LCL-Imp formalize properties of the Equivalential Calculus and the Implication and Falsum Calculus, respectively. Both are subsystems of the classical propositional calculus, restricting the set of allowed connectives to $\{\equiv\}$ and $\{\implies, \bot\}$. For both subsystems, the appropriate variant of the *condensed detachment* inference rule constitutes a *strongly complete* inference system, i.e., whenever a formula semantically follows from a set of premises, it also follows from the set syntactically.

$$A, A \equiv B \vdash B \tag{1}$$

$$A, A \implies B \vdash B \tag{2}$$

A number of complete axiomatizations of both the Equivalential Calculus and the Implication and Falsum Calculus exist and the theorems in our datasets establish connections between them.

## E  CURRICULUM LEARNING VS. SUPERVISED LEARNING

Table 9: *Curriculum Learning vs Supervised Learning trained on proofs with extra steps added for distraction.* FLoP *is barely affected, while supervised learning's performance degrades. Numbers with* $^\star$ *are averaged from 2 runs.*

| Dataset | Proof Lengths | Supervised Succ. | Curriculum Succ. |
|---------|---------------|------------------|------------------|
| RA-1    | 5, 9          | 0.98(0.04)       | **1 (0.01)**     |
|         | 9, 11         | 0.52(0.08)       | **0.98 (0.01)**  |
| RA-2    | 5, 9, 23      | **0.85 (0.04)**  | 0.76(0.02)$^\star$ |
|         | 9, 11, 25     | 0.59(0.08)       | **0.76 (0.01)**$^\star$ |

When training proofs are available, a natural baseline of curriculum learning is supervised learning on the proof steps. While such behavioral cloning sometimes leads to great performance, we show in Table 9 that it greatly depends on the quality of the given proof. We select two sets of training problems for RA-1 and RA-2:

1. **RA-1** $1 + 1 = 2, 1 \cdot 1 = 1$
2. **RA-2** $1 + 1 = 2, 1 \cdot 1 = 1, 1 \cdot 1 \cdot 1 = 1$

We take the "nice" proofs (5, 9 and 23 steps) of these problems and construct variants with 2-3 extra steps added. We observe that supervised learning degrades as superfluous steps are introduced, while FLoP's exploration allows the system to recover and find the original proofs.

## F  FEATURES IN FLoP

FLoP represents states and actions based on previously developed features Kaliszyk et al. (2015a). The features include (suitably hashed) triples, pairs, and singletons of adjacent nodes in the formula trees and the partial proof trees, as well as some global features: number of open goals, number of symbols in them, their maximum size and depth, length of the current path, and two most frequent symbols in open goals. This means that the proof states and the actions are presented as (sparse) fixed-length vectors.

## G  EXPERIMENT HYPERPARAMETERS

Our hyperparameters were selected using small grid searches. We checked standard RL parameters (e.g., the discount factor), parameters related to curriculum scheduling (e.g., local vs. global), neural network architectures (1–5 layers with 128–1024 neurons), feature sizes (64–1024) and training steps ($10^5 - 10^8$). Parameters used in the experiments are described in configuration files which are accessible along with the shared codebase.

## H  EAGER EVALUATION BASED ON THE VALUE MODEL

Given that the evaluated RL algorithms train both a policy and a value model, an alternative of policy-based eager evaluation is to select the action whose successor state has the highest value score. Table 10 shows, however, that the value-based evaluation is much worse for each dataset. We conjecture that this is because assigning a value to a never observed state is much harder than selecting from a smaller set of actions.

Table 10: *Comparing* plCoP *using MCTS and two different eager evaluation methods based on hte policy and value models*

| Dataset | Prover | MCTS | Eager Policy | Eager Value |
|---------|--------|------|--------------|-------------|
| LCL-Eq  | plCoP  | **47%** | 5%        | 1%          |
| LCL-Imp | plCoP  | **61%** | 5%        | 1%          |
| RA-1    | plCoP  | 65%  | **82%**      | 3%          |
| RA-2    | plCoP  | 48%  | **49%**      | 5%          |

# I FAILURE MODES

Despite the apparent simplicity of our arithmetic learning environments, a learning system aiming to solve them has to overcome some hard challenges. We have decided to describe these challenges in detail as they are present in other domains as well, even if it may be harder to detect.

**Failure type 1.** The reward mechanism of our RL system is biased towards shorter proofs. However, many problems have "shortcuts" that allow for shorter proofs, but that do not generalize well. Consider formula $(1+1) + (2 \cdot 2) = (0+2) + 4$. There are two ways to prove this equality: 1) compute the values of the expressions on both sides of the equation and notice that they are the same or 2) show that $1 + 1 = 0 + 2$ and $2 \cdot 2 = 4$. The former generalizes better, but the latter results in a shorter proof. Hence, training on this problem might negatively affect the performance of the prover. This is what prevents FLoP to bootstrap itself in RA-3, i.e., train on easy problems and generalize to harder ones. We find that providing some of the harder problems (having longer proofs) helps to avoid misleading shortcuts.

**Failure mode 2.** fCoP features do not take into account the order of the arguments of a function, i.e., $f(a, b)$ and $f(b, a)$ have the same features. This is problematic for RA-3, since $A = B$ and $B = A$ require different inferences. We addressed this problem by 1) extending state features with those of the preceding action as a substitute of memory, 2) modified the features to include argument order.

**Failure mode 3.** Some "rare" events are hard to generalize because the system sees very few relevant samples during training. This is the case with applying commutativity of equality (replacing $A = B$ with $B = A$), which is only required in RA-3 and ideally only once per proof when we move the focus from one side of the equation to the other. In Experiment 4, when we trained on a single longer proof, we have noticed that the system was very unsure about this action which resulted in many failed proof attempts. Adding another training proof was enough to overcome this and the success score increased from 32% to 67%.

# J THE LEANCOP CONNECTION TABLEAU CALCULUS

FLoP provides guidance for of the very compact leanCoP (Otten & Bibel, 2003) connection tableau calculus. The calculus was originally implemented in Prolog, but it also has an OCaml reimplementation fCoP (Kaliszyk et al., 2015b) and FLoP can be used to guide both systems.

We briefly describe the connection tableau calculus, assuming basic first-order logic and theorem proving terminology (Robinson & Voronkov, 2001). The input is a (mathematical) problem consisting of *axioms* and *conjectures* formally stated in first-order logic (FOL). The calculus searches for *refutational proofs*, i.e. proofs showing that the axioms together with the negated conjectures are *unsatisfiable*. The FOL formulas are first translated to *clause normal form* (CNF), producing a set of first-order *clauses* consisting of *literals*, e.g. $\{\forall X, Y : (f(X)|r(X, Y|\neg f(Y)), f(a)\}$. Proof search starts with a *start clause* as a *goal* and proceeds by building a connection tableau by repeatedly applying *extension steps* and *reduction steps*.

The extension step connects (*unifies*) the *current goal* with a complementary literal of a new clause. This extends the *current branch*, possibly splitting it into several branches if there are more literals in the new clause, and possibly *instantiating* some variables in the tableau. The reduction step connects the current goal to a complementary literal of the *active path*, thus *closing* the current branch. The proof is finished when all branches are closed. The extension and reduction steps are nondeterministic, requiring backtracking in the standard connection calculus. Brute force search such as *iterative deepening* can be used to ensure completeness. Figure 5 shows a *closed connection tableau*, i.e., a finished proof tree where every branch contains *complementary literals* (literals with opposite polarity). This shows that the set of clauses is unsatisfiable.

leanCoP represents theorem proving as a one-person game. The game ends with success if a proof is found. The prover has many choices to make along the way, in particular it can select from several valid extension and reduction steps. Whether a step is valid depends on the unification condition, i.e., if the current goal unifies with the negation of a literal in the corresponding clause. The full information about the game state consists of all previous proof steps, the partial proof tree (proof state) and the current goal.

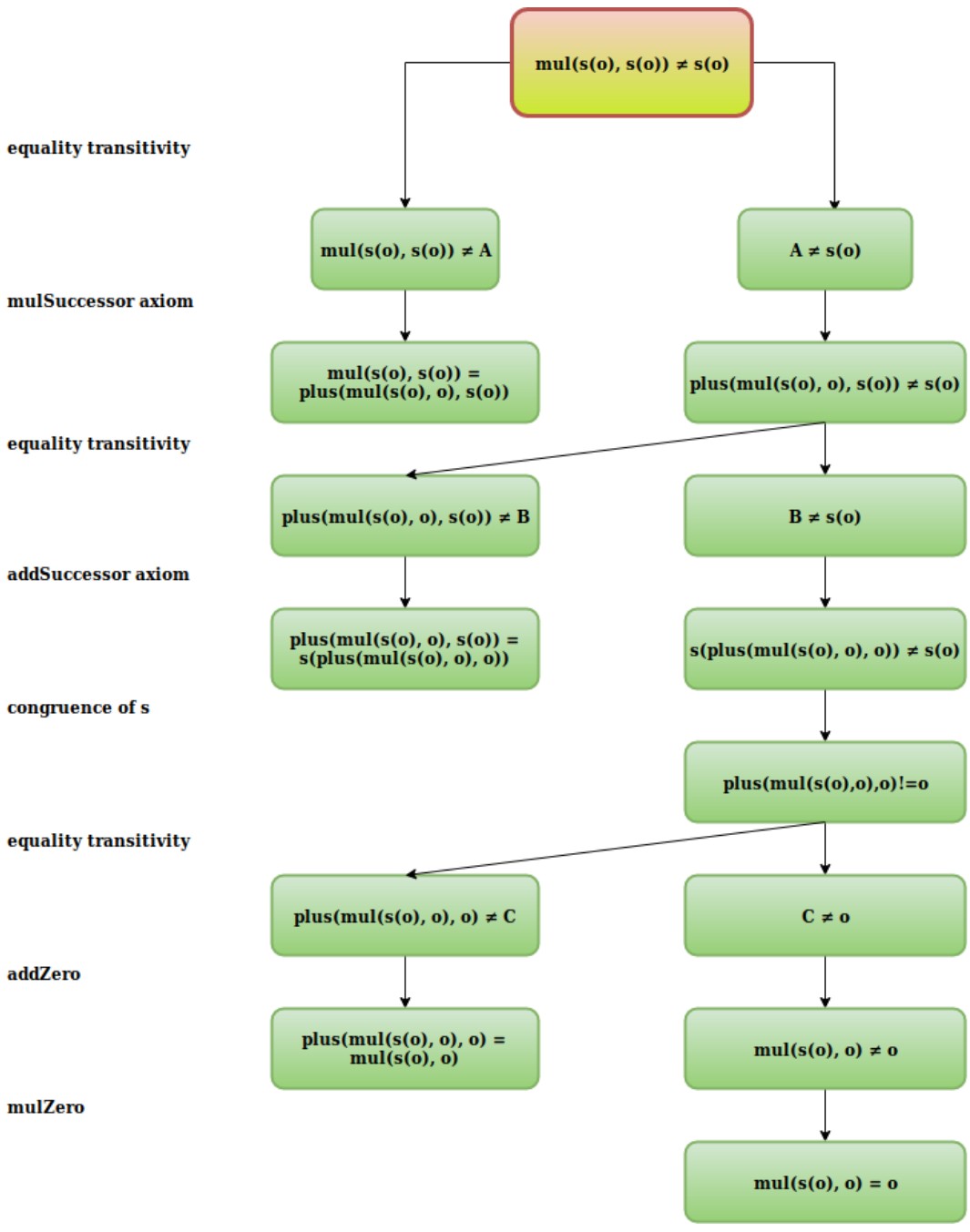

Figure 5: *A closed tableau tree of the proof of* $1 \cdot 1 = 1$. *On the left we list the actions taken in the proof. See* http://bit.ly/ site_atpcurr *for details.*

The search space of the prover is exponentially large in the length of the proof. In leanCoP, the action space is roughly correlated with the size of the axiom set. While this can be large for large problems, typically only a few actions are available in any particular state.

