# OpenReview forum: "Towards Finding Longer Proofs"
_ICLR.cc/2021/Conference — Reject_

### Official Review · AnonReviewer3 · 2020-10-26
**Does reinforcement learning include reasoning by analogy?**

**Rating:** 8
**Confidence:** 3

**Review:**

This work introduces a new algorithm FLoP for theorem proving using reinforcement learning, and tests it on a new evaluation dataset.  FLoP gives direction to a tableau based theorem prover by learning a state machine using curriculum learning applied to a prototype proof.  The authors state that this RL technique has not been previously applied to theorem proving.  I cannot judge this statement but if true it would seem enough novelty to justify publication.   They find that the technique works best on highly structured problems such as proving simple arithmetic statements in unary or binary arithmetic, and less well for problems which benefit from searching through databases of heterogeneous statements.

One could debate whether this deserves the name of "reasoning by analogy".   I suspect it should be called "reasoning by imitation".   To my mind, the term analogy suggests a reasoning process in which some features are extracted from the proof, and then the proof strategies which work for these feature values are selected out of a large set of possibilities including many with different feature values.  I quote from the authors' description at the beginning of section 6 of what they show: "In this highly structured dataset FLoP is capable of extracting a general proof pattern from one or two proofs and successfully generalizing to related proofs of arbitrary length."  This does not sound like analogy as I defined it, rather it sounds like imitating the prototype.

Still, since the comparison with other techniques is encouraging, and since the paper is clearly written and gives a very extensive survey of comparable works, I found it enlightening and would recommend to accept it.

---

> ### Author Response · Authors · 2020-11-17
> **Response to Reviewer #3**
>
> Dear Reviewer #3,
>
> Thank you for your evaluation. With respect to connection with analogy, please see the joint response to all the reviews.
>
> A very natural continuation of our work is your proposed formalisation of proof by analogy in which proofs are considered as proper entities and we learn their features. A model that outputs an entire proof - or maybe first part of a proof - in a single step could be a logical next step, which is the subject of a separate experiment we are currently working on. In the paper, we hope to have shown that we can train models that achieve the kind of confidence in internalising an entire proof pattern required for such systems.

---

### Official Review · AnonReviewer2 · 2020-10-28
**Impressively scalable search with fairly standard RL, possibly tenuous connection to analogy**

**Rating:** 6
**Confidence:** 2

**Review:**

Summary:
This work introduces a method for learning to prove theorems which can leverage prior proving experience in order to discover very long proofs. At its core it works by inputting a corpus of training problems (which can also be annotated with solutions i.e. proofs), training a policy to solve these training problems by curriculum learning. The curriculum works by first supervising on the trace of an entire solution, and then once the system can solve a particular problem, decreasing the amount of the trace that it supervises on. The authors claim that this is a kind of analogical reasoning, because the system's policy is implicitly learning to represent the state/action space on the basis of prior experience.

I'm not convinced that this looks very much like the analogical reasoning advertised in the introduction to the paper , which claims "Reasoning by analogy involves observing the proof of one problem, extracting the core idea, andsuccessfully applying it to another". The system does not work by fetching previous proofs and massaging them into a proof for a new problem, except implicitly through the prior knowledge represented in the weights of the policy. My only real objection to this work is that it doesn't function as advertised. In particular I'm not sure why the learned policy wouldn't be subject to the "the exponential blowup [which causes] the search... to fail beyond a certain depth," which afflicts prior methods.

IMO this paper should be accepted iff this approach is a new advance for automated theorem proving, but should not be accepted on the basis of it's connection to reasoning by analogy. However, it is possible that I have misunderstood how the algorithm works, and would like to be corrected by the authors if indeed the algorithm works as advertised. Because I am not well steeped in the automated theorem proving literature, I do not feel qualified to make the call on how big of an advance this is for that community. However, it does not seem like a large advance for deep learning or for combinatorial search.

Pros:
+ seems to scale to some extraordinarily long proofs! For example, their system can discover proofs with steps on the order of 10^3-10^4. Given the combinatorial explosion in the search space (scales exponentially with proof length) this is quite impressive.
+ reasoning by analogy is crucial yet underexplored. It should be especially important for few-shot learning of proof strategies, and I'm glad that the authors have taken up this line of research.

Cons/questions:
- How sensitive is the system to the training problems from which it forms analogies? For instance experiment 3 hinges on two particular seed expressions--how did you choose them?
- How much do you depend on the particular feature space? Appendix F lists what seem like a relatively impoverished feature representation for the state space. Is this intentional, whereby narrowing the feature space you improve transfer between the target (training) and source (test) proofs? In that case it would seem that the method would struggle absent careful feature engineering, which could limit its applicability beyond the simple theories considered here (nb: although the theories are simple the proofs are extraordinarily long)

Minor concerns:
typo page 3 - "allows for potential reusing" (potential to potentially)
missing citation page 3 - "Despite the unquestioned role of..." (include citation to "Mathematics and Plausible Reasoning")
make figure one larger
figures 2/3: surely you must be doing some kind of kernel density smoothing here?
experiment 1/table 2: describe E2--which is at ceiling--as exploiting hand-crafted domain-specific strategies for arithmetic, preferably in the caption. Otherwise it is confusing why it is at ceiling just by looking at the table

---

> ### Author Response · Authors · 2020-11-17
> **Response to Reviewer #2**
>
> Dear Reviewer #2,
>
> Thank you for your evaluation. With respect to connection with analogy, please see the joint response to all the reviews.
>
> #### Why the learned policy wouldn't be subject to the the exponential blowup?
>
> Search and exponential blowup is likely not avoidable in theorem proving in general. However, mathematics is full of important classes of problems where a deterministic algorithm is achievable and our aim was to demonstrate that FLoP can discover and internalise such algorithms. The models trained on the arithmetic datasets are confident enough to be evaluated without search/backtracking.
>
> #### How sensitive is the system to the training problems from which it forms analogies?
>
> Training problems for Stage 1 and 2 were selected to be simple enough to be solvable by a random policy, while still capturing the complexity of the problem class. Many alternative training problems work just as well. Stage 3 is significantly more complicated. There, a problem can have many different proofs that are very different in terms of how well one can generalize from them.  Appendix I gives some examples when a shorter proof can be misleading, because it exploits a shortcut that is not always applicable. We found that the harder the training problems, the less likely they are solvable using such shortcuts, hence the more valuable training signal they provide. Our experiments with easier training problems (~shorter than 20 steps to prove) often resulted in massive overfitting even when using hundreds of problems. Hence, we selected training problems to be hard enough to resist overfitting to some shortcut. Note that problems in Stage 3 are versatile enough so that our training problems do not cover all the proof patterns in the test set. Many alternative training problems of the same complexity (over 100 steps to prove) can be considered and we conjecture that we would get better results if we trained on more such problems.
>
> #### How much do you depend on the particular feature space?
>
> We decided to use a feature representation that was successfully applied in several previous works in theorem proving, without modification to make our system better comparable. The features provide a fast baseline that were not tuned for our datasets - we used them as provided by the independently developed fCoP kernel. Learned feature extraction of logical objects is a separate line of research and we agree that a natural improvement on FLoP can be to incorporate such methods.  Note, however, that neural formula embedding entails further slowdown of the guidance, making it hard to remain competitive in real time. Several experiments have shown the challenge to improve upon the features used in FLoP by neural embeddings (e.g. https://link.springer.com/chapter/10.1007/978-3-030-29436-6_12 and https://arxiv.org/abs/1911.02065v3).
>
> #### Minor comments
>
> You are right about Figures 2/3, we use kernel density estimation. We will state this more explicitly in the paper. We are grateful for all the other comments and will update the paper accordingly.

---

### Official Review · AnonReviewer1 · 2020-11-02
**How does FLoP relate to reasoning by analogy?**

**Rating:** 4
**Confidence:** 3

**Review:**

This paper proposes a theorem prover based on Proximal Policy Optimization for the connection tableau calculus. This prover is applied on five domain-specific datasets, where theorems are relatively simple but their proofs are long and repetitive. The proposed theorem prover could achieve competitive performance with strong baseline provers, yet requires much few searches.


I think the assumption of this paper is correct that we need more accurate heuristics but not more searches to find longer proofs. The main issue is that it is unclear what is the novelty of the proposed approach. The approach section of the main paper is quite short, just one paragraph (section 3) and one algorithm. It seems that the main approach is to train the theorem prover by reinforcement learning following a specific learning curriculum. It is not mentioned that why the proposed approach has advantages to finding longer proofs, and how the proposed approach is related to reasoning by analogy. Currently, the reasoning by analogy approach mentioned in section 2 seems irrelevant to the proposed approach, except that they may share the same effect and target to reduce the number of searches within the prover.

I think the proposed approach shares the same form of heuristic as the prior work on neural theorem proving, that building a reactive policy to select the next action based on the current proof state. I can not see how this is related to reasoning by analogy, which is described as "Reasoning by analogy involves observing the proof of one problem, extracting the core idea, and
successfully applying it to another". All should we consider all neural-based provers as reasoning by analogy, since they are trained with existing proofs?

=================================================
After reading the responses from the authors and other review comments, I maintain my previous rating of this paper. I am not convinced that the proposed approach is a simple form of analogy reasoning. Trying to build a relationship between the proposed approach and analogy reasoning is uninformative and misleading.

---

> ### Author Response · Authors · 2020-11-17
> **Response to Reviewer #1**
>
> Dear Reviewer #1,
>
> Thank you for your evaluation. With respect to connection with analogy, please see the joint response to all the reviews.
>
> #### Novelty of the proposed approach
>
> We use fairly standard RL techniques, some of which have not yet been applied to theorem proving. In particular, we use continuous, online learning from rollouts coming directly from the policy. To the best of our knowledge, learning from rollouts has not been applied to theorem proving before the first release of FLoP. Since then, there is at least one other paper using policy gradient for theorem proving: https://arxiv.org/abs/1911.02065v3. This learning setup, coupled with curriculum learning, allows us to train on some rather long proofs in Stage 3, with much less overfitting than in supervised learning. We argue that the selected techniques are a good choice if one wants to explore the space around a particular problem (or a particular proof) to such an extent so that it generalises to problems with extreme lengths.

---

### Author Response · Authors · 2020-11-17
**Response to all three reviewers**

Dear Reviewers,

We are greatful for your thorough evaluation. We would like to use this space to answer issues raised by all of you and then respond to individual comments separately.

#### Connection with analogy

In our paper, we have interpreted analogical reasoning as building a model that internalizes a proof, and then successfully applies it to a class of related problems, without relying much on search. The trained model is supposed to "know" the proof of an unseen, yet familiar problem. This is a highly simplified approach which does not capture the full potential of analogy, but we argue that it is a meaningful start that will hopefully lead to more refined solutions. In particular, our model does not yet manipulate on the level of abstraction of proof objects, rather that of a sequence of proof steps. Note, however, that this is also true of previous works on theorem proving by analogy, trying to establish direct matchings between steps in one proof and steps in another, using all sorts of methods/heuristics. Machine learning can make this proof step matching process automatic and allows to replace one-to-one mappings with many-to-many mappings. To some extent, this is true of most works using ML for ATP. Our paper has tried to push this direction further, making analogy and elimination of search more explicit. Prior and parallel works have built similar guiding policies, but they have not demonstrated similarly successful internalisation of a proof method. We will make our interpretation of analogy more explicit in the paper.

---

### Decision · Program_Chairs · 2021-01-07
**Final Decision**

**Decision:**

Reject

**Comment:**

This paper describes an application of reinforcement learning to theorem proving in the connection tableau calculus.  The paper does a reasonable job in the application of RL techniques and the high level issues are important.  However, as the reviewers note, there is little connection to the notion of "analogy" outside of the very general idea that RL methods learn to generalize to novel situations.

I did not find the methods very original as it seems a somewhat mechanical application of RL methods.  That would be fine if the empirical results were convincing or surprising.  However, I found the Robinson arithmetic domains not very interesting as the problems were literally arithmetic, as in 2+5 = 7, rather than theorems such as the commutativity of addition.  The empirical results were not as convincing in the TPTP domains where MCTS seemed to dominate.

Also there are related papers in the area of deep learning applied to theorem proving that I believe dominate this paper ("learning to reason in large theories" and "an inequality benchmark".